# OpenReview forum: "Utility-Diversity Aware Online Batch Selection for LLM Supervised Fine-tuning"
_ICML.cc/2026/Conference — ICML 2026 regular_

### Official Review · Reviewer_TsoH · 2026-03-10

**Soundness:** 3
**Presentation:** 2
**Significance:** 3
**Originality:** 2
**Overall Recommendation:** 5
**Confidence:** 3

**Summary:**

This paper proposes UDS (Utility-Diversity Sampling), an online batch selection methods for fine-tuning LLM.
It uses the nuclear norm---the sum of the singular values of the predicted logits matrix for datapoints during forward process ---to reflect properties of the data samples as an importance score.
The logits matrices of historical samples are projected and accumulated during this process, while the new data points compute the distance as the diveristy score.
Both importance score and diversity score are weight averged into the final metric to evaluate the quality of the data sample.
UDS then select the top-K samples to backpropagate and udpate the LLM.

This paper also presents a lemma and a theorem to show that 1. for a given matrix, its Frobenius norm is less than or equal to its nuclear norm. and the nuclear norm is bounded by square root root of the minimum of the sequence length and vocabulary size.
2. Use two random matrixes to project high-dim logits matrix into low-dim vectors. The vectors are very likely satisfies the Johnson-Lindenstrauss lemma --- pairwise euclidean distances are preserved.

This paper conducts experiments across various datasets, MMLU, Sci-QA, GSM8K and HumanEval. It compares against baselines like MaxLoss, RHO-Loss, etc, on two models llama3.1-8B and Qwen2.5-7B. UDS achieves consistanly 1 or 2 points higher on each tasks (except the 5 for MMLU).

**Compliance With Llm Reviewing Policy:**

Affirmed.

**Final Justification:**

I decide to keep my postitive score.

**Key Questions For Authors:**

1. What are the extra overhead does UDS introduce in flops? the throughput comparsion in table 3 is not very informative for this question because of the partial back prop of UDS (only K samples does the BP process). A FLOPs analysis is very necessary for UDS.

**Limitations:**

yes

**Strengths And Weaknesses:**

**Strengths**
1. UDS proposes a novel perspective by using the internal matrix property (nuclear norm) to guide the model training. The two random matrix projection is an intersting design.
2. This paper conducts extensive experiments, including 6 baseliens, 4 tasks and 2 models. And have servral insightful ablation study.
3. The analysis is solid and informative.

**Weaknesses**
There are a few unclear points regarding the presentation:
1. In figure-2, the p-value should be metioned in the caption. Otherwise the dual Y-axes are confusing.
2. In sec 4.1, the implementatinon Details, the batch size to 8 is ambiguous, the author should explcitly mention it is micro batch size (or device batch size), not the global/effective batch size.
3. Table3 misses the base (before SFT) model performance.

---

> ### Author Rebuttal · Authors · 2026-03-31
>
> *We thank the reviewer for the valuable feedback and believe the responses below sufficiently address the concerns.*
>
> ---
>
> > **W1&W2: Regarding language description**
>
> We thank the reviewer for the advice and will modify the description in the final version. For Figure 2, we will change each subfigure's caption to "Trajectory of Spearman Rank Correlation and Corresponding p-value". For Section 4.1, we will explicitly describe $B$ as the micro batch size.
>
> > **W3: Base model performance in Table 3**
>
> As suggested by the reviewer, we provide the base model performance for Llama-3.1-8B and Qwen-2.5-7B in Table 3 using the same evaluation prompts. The results are listed in **Supplementary Table 8**, where all metrics are lower than those of any fine-tuned baseline in Table 3, as expected. We will add these new results in Table 3.
>
> **Supplementary Table 8: Base model performance in Table 3.**
> |Model|MMLU|ScienceQA|GSM8K|HumanEval|
> |:---:|:---:|:---:|:---:|:---:|
> |Llama-3.1-8B|31.23±0.35|40.15±0.43|30.48±0.74|12.40±1.84|
> |Qwen-2.5-7B|20.05±0.08|80.75±0.04|74.61±0.26|16.48±2.17|
>
> > **Q1: FLOPs breakdown analysis of UDS**
>
> We provide a detailed FLOPs breakdown for different components of UDS across 4 datasets in **Supplementary Table 9**. In a single batch, UDS can be divided into three parts: (1) forward pass for all $B$ samples to obtain logit matrices, (2) computing $s_\text{intra}$ and $s_\text{inter}$ for selection, and (3) partial backpropagation for $K$ out of $B$ samples. We can see that the core selection component of UDS introduces only approximately 1% overhead in terms of FLOPs. We will add these new results and analysis in Appendix C.
>
> **Supplementary Table 9: FLOPs breakdown during the training process for UDS components in Table 3.**
> |Model|Stage|MMLU|ScienceQA|GSM8K|CodeAlpaca|
> |:---:|:---:|:---:|:---:|:---:|:---:|
> |Llama-3.1-8B|forward|3.5e+17|1.1e+17|1.3e+16|4.0e+16|
> ||selection|9.2e+15|1.4e+15|2.1e+14|5.7e+14
> ||backward|3.5e+17|1.1e+17|1.3e+16|4.0e+16|
> |Qwen-2.5-7B|forward|3.4e+17|1.0e+17|1.4e+16|3.9e+16|
> ||selection|1.1e+16|1.7e+15|3.0e+14|6.8e+14|
> ||backward|8.5e+16|1.0e+17|7.1e+15|3.9e+16|
>
> ---
> *Finally, we express sincere gratitude for your examination and your suggestion help improve quality of this work a lot.*

---

> > ### Author Rebuttal · Reviewer_TsoH · 2026-04-02
> >
> > Thanks to the author for the further explanation; all my questions have now been answered.

---

> > > ### Author Response · Authors · 2026-04-02
> > >
> > > We are delighted that our response addressed your concerns. We sincerely appreciate your kind support and the AC's effort in coordinating this review, and look forward to receiving your favorable rating. We will incorporate these discussions and related new results into our final version.

---

### Official Review · Reviewer_f81R · 2026-03-12

**Soundness:** 3
**Presentation:** 3
**Significance:** 2
**Originality:** 3
**Overall Recommendation:** 4
**Confidence:** 3

**Summary:**

This paper proposes a data selection method for SFT without relying on external resources. The proposed approach scores candidate samples using two components: an intra-sample score and an inter-sample diversity score. The intra-sample score is computed using the nuclear norm of the logits matrix, which the authors argue captures both optimization utility and intra-sample diversity. The inter-sample score measures the distance between the current sample and recently selected samples in a low-dimensional projected representation

**Compliance With Llm Reviewing Policy:**

Affirmed.

**Final Justification:**

I have increased the score. While the authors’ heuristic explanation is reasonable, I remain concerned that the core scoring rule is only partially justified. Consequently, I am not fully convinced by the paper’s main findings.

**Key Questions For Authors:**

1. How does UDS behave in the presence of label noise, outliers or adversarial samples? Could the proposed scoring mechanism inadvertently prioritize such samples due to high logit magnitude or diversity?

2. While the paper provides empirical evidence, can the authors further clarify why nuclear norm is a particularly suitable proxy for optimization utility compared with simpler alternatives (loss-based, entropy-based, gradient norm, Frobenius norm scoring)?

3. The author said "Due to model complexity and nonlinearity, strictly proving the nuclear norm’s linear correlation with loss reduction is challenging." However, I think there is no clear correlation, let alone a linear correlation. Therefore, I am concerned about the rigor of this description. In addition, the loss depends on the ground truth, but the author tries to claim a correlation between the loss and the nuclear norm without information about the ground truth.

**Limitations:**

yes

**Strengths And Weaknesses:**

Strength:

1. The proposed approach aims to reduce training cost while maintaining or improving model performance. While its general applicability remains uncertain, this topic holds significant value for real-world training pipelines.

2. The proposed UDS framework is conceptually simple and easy to understand. The method decomposes the sample scoring mechanism into two intuitive components.

Weakness:

1. The paper argues that the nuclear norm of the logits matrix captures both optimization utility and intra-sample diversity. While the paper provides intuition and empirical correlations, the theoretical connection between nuclear norm and actual optimization utility remains unclear.

2. In particular, the analysis based on the first-order Taylor expansion only loosely motivates why larger logit magnitudes may correlate with larger loss reduction, but it does not formally justify why the nuclear norm should serve as a reliable proxy for optimization utility across different models, tasks, or training stages. Most of the theoretical intuitions used in the paper are well known in the literature (e.g., relations between nuclear norm, Frobenius norm, and matrix rank), although they are not explicitly connected to the SFT setting.

3. This data selection method prioritizes samples with large nuclear norms and high diversity relative to recently selected samples. However, such criteria may also assign high scores to noisy, mislabeled, or adversarial samples that produce unusual logits structures or strong prediction conflicts.

4. The proposed online batch selection method requires forwarding all samples to compute their scores (e.g., nuclear norm and diversity) before selecting the subset. But forward passes already constitute a substantial portion of LLM training.

---

> ### Author Rebuttal · Authors · 2026-03-31
>
> *We thank the reviewer for the constructive feedback and believe our responses below can address the reviewer's concerns.*
>
> ---
>
> > **W1&W2&Q3: Theoretical justification of nuclear norm**
>
> **(1) Empirical Validation of Mathematical Intuitions**: Our study starts from sound mathematical analysis, connecting the Taylor expansion of loss reduction to matrix norms, and validates these intuitions through rigorous correlation analysis. The strong empirical evidence (Spearman ~0.8 in Figure 2) demonstrates the nuclear norm as a reliable proxy for online data selection.
>
> **(2) Theoretical Demonstration as a Recognized Limitation**: We acknowledge that rigorously proving the exact theoretical connection is highly challenging due to LLM complexity and non-linearity, and leave this as a limitation and future direction.
>
> **(3) Alignment with Standard Field Practices**: Empirically guided method design is widely adopted in machine learning. Previous studies leveraged the nuclear norm for unsupervised accuracy estimation showing strong linear correlations (>0.95) without labels [1], and utilized Diff-eRank as an unsupervised proxy for loss reduction [2].
>
> We'll add this to Appendix F:
> > Though our method relies on mathematical intuition and strong empirical correlations (Spearman ~0.8) for effective data selection, a formal theoretical proof remains open given LLM complexity. Future work will develop a theoretical framework bridging matrix norms and loss reduction in LLM SFT.
>
> > **W2: Contribution when using math tools**
>
> While the foundational mathematical relations are established in linear algebra, as the reviewer notes, they have not been connected to the LLM SFT setting. Our contribution lies in bridging this gap: we translate static matrix properties into a dynamic, compute-efficient solution for online data selection, carefully adapting these intuitions as a practical proxy for loss reduction that balances data utility and diversity without prohibitive computational overhead.
>
> > **W3&Q1: The impact of outlier samples on UDS**
>
> As discussed in Appendix B.1 and in [3], online batch selection is not applied in isolation but forms a complementary pipeline with offline data selection: offline selection filters out outliers and adversarial examples, while online selection captures dynamic shifts in data importance during training.
>
> In the curated SFT datasets used in our experiments, noisy or adversarial samples do not exist. To further test robustness, we inject 5% random label noise into the MMLU training set (**Supplementary Table 7**). All baselines degrade, but UDS still achieves the best performance. Moreover, only 2% of the mislabeled samples are selected by UDS, far below the 12.5% expected under uniform selection. We will add the new results in Appendix C.
>
> **Supplementary Table 7: Accuracy Comparison on noisy datasets using Qwen-2.5-7B.**
> |Corruption Ratio|Regular|Random|MaxLoss|MaxGrad|RHO-Loss|GREATS|UDS|
> |:-:|:-:|:-:|:-:|:-:|:-:|:-:|:-:|
> |0%|55.32±0.79|54.26±1.85|54.51±1.37|54.33±0.69|57.08±0.74|58.19±0.49|63.34±0.36|
> |5%|53.89±0.93|52.58±2.14|52.84±1.56|52.71±0.82|55.64±0.91|56.87±0.63|62.18±0.42|
>
> > **W4: Forward pass already constitutes overhead**
>
> The forward pass is a shared minimum requirement across all online batch selection methods to capture the current model state and guide selection. Beyond the forward pass, the additional overhead of UDS is negligible (see **Supplementary Table 9** in our response to Reviewer TsoH). We believe this is an inherent property of online batch selection rather than a specific weakness of UDS.
>
> > **Q2: Advantages of the nuclear norm in terms of optimization utility compared to alternatives**
>
> We compare UDS against each alternative below:
> - **MaxLoss:** Captures current difficulty but not loss reduction potential. As shown in Table 2, high-loss samples may be "too hard" and unlearnable, yielding low optimization utility.
> - **Entropy:** Only captures intra-sample diversity and lacks a direct link to optimization utility.
> - **MaxGrad:** Captures optimization utility but requires full backpropagation, making it significantly slower than other methods (Table 3).
> - **Frobenius norm:** Correlates with loss reduction (Figure 2c) but only captures magnitude without the rank structure that reflects intra-sample diversity.
>
> In contrast, nuclear norm uniquely captures both optimization utility (Figure 2a) and intra-sample diversity (Figure 2b) from a single forward pass.
>
> **References**
>
> [1] Confidence and Dispersity Speak: Characterising Prediction Matrix for Unsupervised Accuracy Estimation. ICML2023.
>
> [2] Diff-eRank: A Novel Rank-Based Metric for Evaluating Large Language Models. NeurIPS2024.
>
> [3] GREATS: Online Selection of High-Quality Data for LLM Training in Every Iteration. NeurIPS2024.
>
> ---
> *Finally, thanks again for your time and helpful comments. We hope all concerns are well addressed. It would be appreciated if you can reconsider the evaluation of our work.*

---

> > ### Author Rebuttal · Reviewer_f81R · 2026-04-03
> >
> > Thank you for the rebuttal. I will raise my score. I think the authors’ heuristic explanation is reasonable; however, I remain concerned that the core scoring rule is only partly justified. As a result, I am not fully convinced of the paper’s main observations.

---

> > > ### Author Response · Authors · 2026-04-04
> > >
> > > We thank **Reviewer f81R** for raising the score and acknowledging the reasonability and contribution of this research. We agree that further theoretical investigation of the logits' nuclear norm's influence towards loss reduction is an open challenge due to the complexity of large models, yet it will bring more insights and practical use into effective online data selection. Hence, we'll continue to dig deeper in the future. Once again, we are grateful to Reviewer f81R and the Area Chair' efforts. All promised modifications will be reflected in the updated manuscript.

---

### Official Review · Reviewer_C8Ac · 2026-03-13

**Soundness:** 2
**Presentation:** 3
**Significance:** 2
**Originality:** 3
**Overall Recommendation:** 4
**Confidence:** 4

**Summary:**

This paper addresses online batch selection for SFT of LLMs. The authors identify three key limitations of existing methods: neglecting diversity in favor of utility-only scoring, reliance on external resources such as reference models or validation sets, and extra computational overhead beyond full-dataset training. To address these, they propose UDS, which jointly captures data utility and intra-sample diversity via the nuclear norm of the logits matrix, and estimates inter-sample diversity through low-dimensional embedding comparisons against a lightweight historical sample buffer.

**Compliance With Llm Reviewing Policy:**

Affirmed.

**Key Questions For Authors:**

See the weaknesses.

**Limitations:**

Yes.

**Strengths And Weaknesses:**

Strengths:
1. The core scoring mechanism is technically well-grounded. The use of the nuclear norm as a unified proxy for optimization utility and intra-sample diversity is justified.
2. The paper is clearly written and easy to follow. UDS addresses the online batch selection without requiring external resources such as reference models or validation sets, making it directly deployable.
3. While the nuclear norm and random projections are established techniques, their combination in the context of online batch selection for LLM SFT is well-motivated.

Weaknesses:
1. The selection of the $\alpha$ significantly impacts UDS performance. As shown in Figure 7, $\alpha$ of $2.5 \times 10^{-3}$ on MMLU causes the accuracy to drop to 56, which is notably lower than RHO-Loss and GREATS. This sensitivity suggests that UDS requires intensive hyperparameter tuning for different datasets and models, which prevents UDS from being directly adopted to other tasks.
2.  As shown in C.1, it is unclear whether the inter-sample diversity term α·s_inter remains at a comparable magnitude to s_intra after this scaling, or whether it becomes effectively negligible in practice. The authors should report the typical magnitude of each term after scaling across different datasets and models to confirm that both components meaningfully contribute to the final score. In Table 4, does the "Only Nuclear Norm" and "Only Diversity Distance" variants mean to use s_intra and s_inter independently?
3. Could the authors provide a detailed analysis of why UDS consistently outperforms full fine-tuning while utilizing only a fraction of the data? For example, the authors should analyze whether the samples that were not selected align with the data categorization presented in Table 2.
4. Regarding the comparison with FisherSFT in C.6, could the authors provide more details, such as a comparison of the time used to select data and peak memory consumption for both FisherSFT and UDS during the training process?

---

> ### Author Rebuttal · Authors · 2026-03-31
>
> *We thank the reviewer for the valuable feedback. We believe the responses below sufficiently address all concerns.*
>
> ---
>
> **W1: Sensitivity to hyperparameter ($\alpha$)**
>
> While UDS introduces $\alpha$ as a hyperparameter, baseline methods are also not parameter-free: RHO-Loss requires choices for holdout set construction and the reference model; GREATS depends on a validation set whose size affects performance. For a fair comparison, Table 3 reports the best results after tuning these method-specific hyperparameters for all methods. In fact, UDS requires tuning only a single scalar $\alpha$, which is fewer hyperparameters than both RHO-Loss and GREATS, making it simpler to tune in practice.
>
> **W2: Magnitude of $s_\text{intra}$ and $\alpha \cdot s_\text{inter}$**
>
> We report the average magnitude of both terms for selected samples across training in **Supplementary Table 3**. For each model-dataset combination, both scores are of comparable magnitude, confirming that both components meaningfully contribute to the final score. We will add these new results in Appendix C.
>
> To the reviewer's second question: the answer is yes, "Only Nuclear Norm" and "Only Diversity Distance" in Table 4 use $s_\text{intra}$ and $s_\text{inter}$ independently.
>
> **Supplementary Table 3: Average magnitude of $s_\text{intra}$ and $\alpha\cdot s_\text{inter}$ under settings in Table 3.**
> |Model|Score|MMLU|ScienceQA|GSM8K|HumanEval|
> |:-:|:-:|:-:|:-:|:-:|:-:|
> |Llama-3.1-8B|$s_\text{intra}$|1.9e+5|6.6e+4|9.6e+4|6.3e+4|
> ||$\alpha\cdot s_\text{inter}$|4.2e+5|4.7e+4|6.9e+4|3.9e+4|
> |Qwen-2.5-7B|$s_\text{intra}$|3.1e+5|8.7e+4|1.7e+5|7.9e+4|
> ||$\alpha\cdot s_\text{inter}$|2.8e+5|4.6e+4|4.6e+5|5.8e+4|
>
> **W3: Why UDS outperforms full fine-tuning with less data**
>
> Good question. We hypothesize that in full-data fine-tuning, a considerable portion of the training budget may be spent on redundant or stage-misaligned samples, which can dilute effective updates and hurt generalization. To verify this, we analyze UDS's selection behavior on Llama-3.1-8B / MMLU (50% selection ratio) following the reviewer's suggestion. Since MMLU is trained for only one epoch, we first run Regular full fine-tuning, record each sample's loss before ($\ell_\text{init}$) and after ($\ell_\text{final}$) training, and classify all samples into the four categories of Table 2 using their respective medians as thresholds. We then run UDS and examine the category distribution among selected and non-selected samples. **Supplementary Table 4** shows the result.
>
> **Supplementary Table 4: Category distribution (%) of selected vs. non-selected samples (Llama-3.1-8B, MMLU).**
> |Category|in Selected|in Non-selected|
> |:-:|:-:|:-:|
> |High Utility (High→Low)|55.7|11.5|
> |Too Easy (Low→Low)|16.9|38.1|
> |Too Hard (High→High)|19.1|34.5|
> |Overfitted (Low→High)|8.3|15.9|
>
> We can see the selected samples are predominantly High Utility, while non-selected samples are disproportionately Too Easy or Too Hard. This confirms that UDS outperforms full fine-tuning by concentrating the training budget on learnable, high-value samples. We will add these results in Appendix C.
>
> **W4: Comparison with FisherSFT**
>
> Since Qwen-2.5-7B's embedding dimension (3584) is much larger than that of the GPT-2 used in the original paper, directly applying FisherSFT causes OOM errors. Thus, we apply a random down-projection (similar to UDS) to 1024 dimensions (the maximum feasible on a single NVIDIA RTX 3090) and refer to this adapted version as FisherSFT below.
>
> For time consumption, FisherSFT is an offline method that performs a single forward pass over the entire dataset and selects data before training, whereas UDS is an online method that performs forward pass and scores samples at every iteration across all epochs. Directly comparing total wall-clock time between these two paradigms is therefore not meaningful. Instead, we compare the algorithm-specific overhead per epoch: for UDS, this is the cumulative time to compute $s_\text{intra} + \alpha \cdot s_\text{inter}$ after the forward pass; for FisherSFT, this is the time for greedy subset selection after embeddings are computed. Results are reported in **Supplementary Table 5**.
>
> For peak GPU memory consumption, we compare both methods under the same hardware setup. Results are reported in **Supplementary Table 6**.
>
> As shown, UDS is consistently more efficient in both time and memory across all datasets. We will add these new results and analysis in Appendix C.6.
>
> **Supplementary Table 5: Algorithm-specific time (wall-clock seconds).**
>
> |Method|MMLU|ScienceQA|GSM8K|HumanEval|
> |:-:|:-:|:-:|:-:|:-:|
> |FisherSFT|635|1672|1281|10107|
> |UDS|584|186|463|645|
>
> **Supplementary Table 6: Peak GPU memory (GB).**
>
> |Method|MMLU|ScienceQA|GSM8K|HumanEval|
> |:-:|:-:|:-:|:-:|:-:|
> |FisherSFT|22.9|23.0|22.0|22.8|
> |UDS|19.2|19.2|18.0|19.4|
>
> ---
> *Finally, we express sincere gratitude for your examination and your suggestion help improve quality of this work a lot.*

---

> > ### Author Rebuttal · Reviewer_C8Ac · 2026-04-03
> >
> > Thanks for your reply, and all my concerns have been addressed.

---

> > > ### Author Response · Authors · 2026-04-04
> > >
> > > We are encouraged by the reviewer’s positive feedback and are delighted that our response has fully addressed the remaining concerns. We extend our sincere gratitude to Reviewer C8Ac and the Area Chair for their efforts and insightful review, which have been instrumental in enhancing our work. We will incorporate all discussed revisions into the final manuscript.

---

### Official Review · Reviewer_8vci · 2026-03-13

**Soundness:** 3
**Presentation:** 3
**Significance:** 3
**Originality:** 3
**Overall Recommendation:** 5
**Confidence:** 2

**Summary:**

The paper proposes Utility-Diversity Sampling (UDS), a online batch selection method for efficient Supervised Fine-Tuning (SFT).
UDS leverages the logits matrix generated during the forward pass. It calculates an intra-sample importance score using the nuclear norm of the logits to capture optimization utility and intra-sample diversity. Additionally, it computes an inter-sample importance score by projecting the logits into a low-dimensional space via a subsampled randomized Fourier transform (SRFT) and comparing it against a memory buffer of historical samples. Experiments show that UDS outperforms the baseline and improves the training efficiency.

**Compliance With Llm Reviewing Policy:**

Affirmed.

**Final Justification:**

All my concerns has been resolved.

**Key Questions For Authors:**

In standard SFT pipelines, it is common to train for multiple epochs (as noted in your configuration for ScienceQA with 20 epochs, and Code Alpaca with 2 epochs). How does the model perform when encountering the exact same data point in subsequent epochs. Correspondingly, how do the intra-sample (nuclear norm) and inter-sample similarity metrics evolve for a specific sample across the entire training lifecycle? Does a high-utility sample eventually decay into a "too easy" state and get naturally filtered out in later epochs? It would significantly strengthen the paper to visualize the trajectory of these metric scores for a few representative samples over multiple epochs.

**Limitations:**

yes

**Strengths And Weaknesses:**

### Strengths
1. Impressive Performance: The proposed UDS framework consistently outperforms the baselines (MaxLoss, MaxGrad) and state-of-the-art online selection baselines (RHO-Loss, GREATS) across multiple benchmarks. It is beneficial to the LLM community.
2. Experimental validation is strong. The paper is well-structured and presentation is clear.

### Weaknesses
1.The scalability to larger models and full-Parameter SFT remains unclear. The core experiments predominantly rely on LoRA fine-tuning (rank=8) and the largest backbones evaluated are Llama-3.1-8B and Qwen-2.5-7B. Although the authors provide a full-parameter tuning experiment in the appendix (Table 11) , it is still restricted to the 7B scale. The scalability and robustness of UDS when applied to larger SOTA model remain underexplored.
2. The scalability to sequence length remains unclear. The overhead of SVD in long-context scenarios maybe unacceptable. The maximum sequence lengths evaluated are relatively short (e.g., 512 in main experiments, and 2048 in the appendix). The effectiveness and system-level efficiency of UDS on truly long-context data (e.g., >32k tokens) are highly questionable.

---

> ### Author Rebuttal · Authors · 2026-03-31
>
> *We thank the reviewer for the constructive comments and provide our response below.*
>
> ---
>
> > **W1: Scalability to larger models**
>
> We agree that evaluating on models beyond 7B would further strengthen the scalability claims of UDS. To address this, we additionally conduct full-parameter fine-tuning on Qwen-2.5-14B following the setting in Table 3. As shown in **Supplementary Table 1**, UDS consistently outperforms all baseline methods, and the overall trend aligns with the observations in Table 3, confirming that UDS scales well to larger models.
>
> We will add these new results in Appendix C.
>
> **Supplementary Table 1: Accuracy Comparison for Different Online Batch Selection Methods with Full Fine-tuning on Qwen-2.5-14B.**
> |Method|MMLU|ScienceQA|GSM8K|HumanEval|
> |:---:|:---:|:---:|:---:|:---:|
> |Regular|63.72±0.51|95.86±0.24|83.66±0.21|49.36±0.33|
> |Random|62.39±0.83|94.91±0.51|83.14±0.27|47.52±0.25|
> |MaxLoss|62.68±0.74|94.85±0.31|83.25±0.41|48.12±0.62|
> |MaxGrad|62.55±0.61|95.12±0.42|83.10±0.38|47.85±0.55|
> |RHO-Loss|64.21±0.45|95.25±0.28|83.85±0.32|48.65±0.48|
> |GREATS|64.85±0.53|95.54±0.35|84.12±0.35|49.05±0.51|
> |**UDS**|**65.86±0.79**|**96.33±0.19**|**84.57±0.26**|**50.30±0.47**|
>
> > **W2: Scalability to longer sequences**
>
> We agree that evaluation under longer contexts would further support the scalability of UDS. Following the setting in Table 13, we additionally test on the MATH dataset with maximum sequence length of 32768. As shown in **Supplementary Table 2**, UDS maintains its advantage under both settings, confirming its robustness in long-context scenarios.
>
> Regarding efficiency, let the sequence length be $N$. The SVD computation has time complexity $O(N^2)$, which may appear costly. However, the attention mechanism also scales as $O(N^2)$ and dominates the forward and backward passes at long sequence lengths. Therefore, the relative overhead of SVD does not grow disproportionately. This is also supported by our empirical measurements showing that the SVD process accounts for only about 2-3% of total training time at both sequence lengths of 2048 and 32768.
>
> We will add these new results and analysis in Appendix C.4.
>
> **Supplementary Table 2: Additional Studies on MATH with Longer CoT Reasoning (max sequence length = 32768). Average accuracy using Qwen-2.5-7B.**
> |Sequence Length|Regular|Random|MaxLoss|MaxGrad|RHO-Loss|GREATS|UDS|
> |:---:|:---:|:---:|:---:|:---:|:---:|:---:|:---:|
> |32768|60.92±0.16|57.81±0.30|57.65±0.45|57.74±0.58|60.38±0.35|60.71±0.27|**61.35±0.32**|
>
> > **Q1: Sample score trajectory analysis**
>
> We thank the reviewer for this suggestion. We conduct a case study on ScienceQA with Llama-3.1-8B (Table 3 setting). Specifically, we select the top-200 samples by joint score ($s_\text{intra}+\alpha\cdot s_\text{inter}$) in the first epoch, then track their metric evolution and selection ratio across all 20 epochs. Since the nuclear norm captures both optimization utility and intra-sample diversity, we additionally report the Frobenius norm to isolate the utility component.
>
> As shown in **Supplementary Figure 1** ([Link](https://anonymous.4open.science/r/figure2-005B/Supplementary%20Figure%201.png)), the Frobenius norm gradually decays, confirming that initially high-utility samples progressively lose optimization utility as the model learns from them, which is consistent with the reviewer's "too easy" hypothesis. However, the nuclear norm increases, indicating that the singular value distribution of logits becomes flatter over training. While the overall logit magnitude shrinks, predictions spread across more tokens rather than concentrating on a few dominant ones, thereby increasing intra-sample diversity. The inter-sample score $s_\text{inter}$ also decreases, primarily because (1) the memory buffer stores more representations over time, reducing nearest-neighbor distances, and (2) continued fine-tuning causes same-domain samples to converge in the projected feature space. These are all as expected.
>
> Finally, the selection ratio of these top samples gradually declines, showing that initially top-ranked samples gradually lose their selection advantage. This demonstrates that UDS naturally adapts its selection behavior across epochs: high-utility samples are prioritized early when most informative and progressively yield to other samples as training converges.
>
> We will add these new results and analysis in Appendix C.
>
> ---
> *Finally, we express sincere gratitude for your examination and your suggestion help improve quality of this work a lot.*

---

> > ### Author Rebuttal · Reviewer_8vci · 2026-04-02
> >
> > Thanks for the rebuttal and additional results. All my concerns has been resolved. I have raised my rate.

---

> > > ### Author Response · Authors · 2026-04-02
> > >
> > > We appreciate the encouraging remarks and are pleased that our response has resolved all concerns. We thank Reviewer 8vci and the Area Chair for their efforts and constructive suggestions, which have significantly improved the quality of our work. All revisions will be incorporated into the updated manuscript.

---

### Decision · Program_Chairs · 2026-04-30

**Decision:**

Accept (regular)

**Comment:**

The paper proposes Utility-Diversity Sampling (UDS), an online batch selection method for efficient LLM supervised fine-tuning. The authors did a good job during the rebuttal, and all reviewers ultimately recommended acceptance. Thus, the AC can easily recommend acceptance. However, the authors are strongly encouraged to include the additional experiments and discussions in the final version.